Evidence synthesis 

health and disease and epidemiology/ immunology/microbiology

vitamin D, COVID-19, immunology

**Author for correspondence:**
Jonathan Rhodes
e-mail: rhodesjm@liverpool.ac.uk

# Vitamin D and COVID-19: evidence and recommendations for supplementation

George Griffin[1], Martin Hewison[2], Julian Hopkin[3], Rose Kenny[4], Richard Quinton[5], Jonathan Rhodes[6], Sreedhar Subramanian[7] and David Thickett[8]

[1]Infectious Diseases and Medicine, St George's University of London, London, UK
[2]Institute of Metabolism and Systems Research, University of Birmingham, Birmingham, UK
[3]Medical School, Swansea University, Swansea, West Glamorgan, UK
[4]Medical Gerontology, Trinity College Dublin School of Medicine, Dublin, Ireland
[5]Endocrinology, Newcastle University Faculty of Medical Sciences, Newcastle upon Tyne, UK
[6]Institute of Translational Medicine, University of Liverpool, Liverpool, UK
[7]Gastroenterology, Royal Liverpool University Hospital, Liverpool, UK
[8]Institute of Inflammation and Ageing, University of Birmingham College of Medical and Dental Sciences, Birmingham, UK

JR, 0000-0002-1302-260X

Vitamin D is a hormone that acts on many genes expressed by immune cells. Evidence linking vitamin D deficiency with COVID-19 severity is circumstantial but considerable—links with ethnicity, obesity, institutionalization; latitude and ultraviolet exposure; increased lung damage in experimental models; associations with COVID-19 severity in hospitalized patients. Vitamin D deficiency is common but readily preventable by supplementation that is very safe and cheap. A target blood level of at least 50 nmol l$^{-1}$, as indicated by the US National Academy of Medicine and by the European Food Safety Authority, is supported by evidence. This would require supplementation with 800 IU/day (not 400 IU/day as currently recommended in UK) to bring most people up to target. Randomized placebo-controlled trials of vitamin D in the community are unlikely to complete until spring 2021—although we note the positive results from Spain of a randomized trial of 25-hydroxyvitamin D3 (25(OH)D3 or calcifediol) in hospitalized patients. We urge UK and other governments to recommend vitamin D supplementation at 800–1000 IU/day for all, making it clear that this is to help optimize immune health and not solely for bone and muscle health. This should be mandated for prescription in care homes, prisons and other institutions where people are likely

to have been indoors for much of the summer. Adults likely to be deficient should consider taking a higher dose, e.g. 4000 IU/day for the first four weeks before reducing to 800 IU–1000 IU/day. People admitted to the hospital with COVID-19 should have their vitamin D status checked and/or supplemented and consideration should be given to testing high-dose calcifediol in the RECOVERY trial. We feel this should be pursued with great urgency. Vitamin D levels in the UK will be falling from October onwards as we head into winter. There seems nothing to lose and potentially much to gain.

# 1. Vitamin D is a hormone that regulates many genes and is dependent on ultraviolet (B) skin exposure for its generation

## 1.1. Vitamin D synthesis and sources

Vitamin D is derived from 7-dehydrocholesterol in the skin by the action of ultraviolet B (UVB, wavelength 280–315 nm) splitting a carbon to carbon bond (C9–C10) thus opening up its B ring—it is structurally related to other cholesterol-derived hormones such as cortisol, testosterone and oestrogen. Formation of the active hormonal form of vitamin D, 1,25 dihydroxyvitamin D ($1,25(OH)_2D$), requires 25-hydroxylation, performed in the liver, and 1-hydroxylation, performed in the kidneys but also in many immune cells and epithelial cells by the action of the enzyme 1-hydroxylase also referred to as CYP27B1 [1].

Vitamin D exists in two forms, D2 and D3. Vitamin D2 (ergocalciferol) derives from UVB irradiation of the yeast and plant sterol ergosterol, and vitamin D3 (cholecalciferol) is found in oily fish and cod liver oil and is also made in the human skin. It is very hard to obtain sufficient vitamin D from food. Oily fish is the only substantial dietary source. Other sources that include liver, eggs and mushrooms (the latter only if they have been UV irradiated) provide only modest amounts of vitamin D. The main source of vitamin D is its generation by the action of UVB on the skin [2].

## 1.2. Seasonal variation in vitamin D

The amount of UVB obtained from sunlight is affected by the height of the sun in the sky (zenith angle) and hence by season and latitude. Ozone absorbs around 95% of UVB. When the sun is low the UVB has to travel further through the ozone layer, increasing absorption or reflection back into space. In the UK sufficient UVB for vitamin D synthesis only occurs on sunny days, from around 10.00 to 15.00, April to September. Wacker & Holick [2] found that one full body UV exposure causing slight skin pinkness is equivalent to an oral intake of 250–625 µg (10 000–25 000 IU) vitamin D3. A model was formulated to help estimate how long an exposure is required. This model suggests that 10–20 min of daily sun exposure during summer months in the UK may achieve an increase of 5–10 nmol $l^{-1}$ in serum 25(OH)D concentration. People living more than 35° latitude away from the equator, including the UK (London, latitude 51.5°), are unlikely to receive sufficient UVB to avoid insufficiency without supplementation. Consequently, vitamin D levels in the UK in February are about 50% of those in September [3]. Pollution, particularly sulfur dioxide, can block UVB substantially. Sunscreen blocks UVB radiation but it is thought unlikely that conventional use of sunscreen causes vitamin D deficiency [4]. People with limited outdoor exposure or who routinely wear more extensive clothing will have reduced vitamin D synthesis and this presumably accounts for the surprisingly high rates of vitamin D deficiency in some countries nearer the equator [5].

## 1.3. Immune cells express CYP27B1 and contain many vitamin D-responsive genes

Many immune cells express the vitamin D receptor (VDR), the receptor for $1,25(OH)_2D$. Moreover, antigen-presenting cells from the innate immune system, such as macrophages or dendritic cells, also express the CYP27B1 enzyme and are, therefore, able to convert 25(OH)D to active $1,25(OH)_2D$ [1,6]. It has been estimated that several hundred genes, including many that are expressed in immune cells, are vitamin D-responsive [7,8].

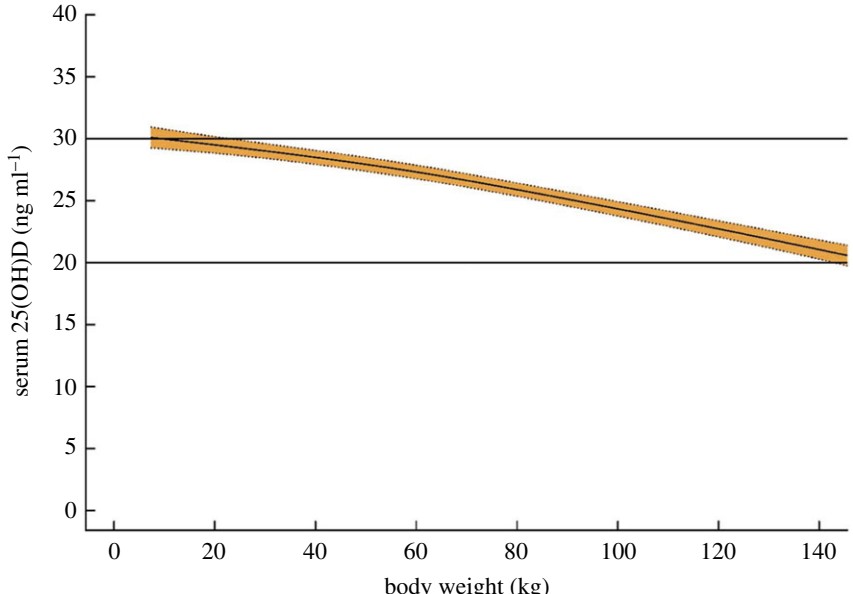

**Figure 1.** Relationship between body weight and serum vitamin D. Estimated mean serum 25-hydroxyvitamin D (25(OH)D) level, with 95% confidence interval, of greater than or equal to 1-year-old US residents ($n = 31\,934$) from the National Health and Nutrition Examination Survey, 2003–2010. The lines at 20 ng ml$^{-1}$ (50 nmol l$^{-1}$) and 30 ng ml$^{-1}$ (75 nmol l$^{-1}$) represent the sufficiency thresholds recommended by the US National Academy of Medicine and the Endocrine Society, respectively. From [16] with permission.

## 2. People with dark skin, who are obese, who are elderly or who are institutionalized are more likely to be vitamin D deficient

### 2.1. Impact of ultraviolet (B) on vitamin D synthesis is reduced in people with darker skin

Light skin colour in Northern Europeans has evolved over the past 75 000 years in order to ensure sufficient vitamin D synthesis to enable the survival of humans at northern latitudes [9]. Nevertheless, the majority of UK white adults do not synthesize sufficient vitamin D from sunlight exposure. Darker-skinned people need even more sun exposure to achieve adequate vitamin D synthesis [2,10]. Herrick *et al.* [11] in a cohort study of 16 180 from the USA showed that the prevalence of vitamin D deficiency (less than 30 nmol l$^{-1}$) was 17.5% in non-Hispanic black compared with 2.1% in non-Hispanic whites. A meta-analysis of 394 cross-sectional studies showed a correlation between vitamin D levels and latitude in Caucasians ($-0.69 \pm 0.30$ nmol l$^{-1}$ per degree, $p = 0.02$), but not for non-Caucasians ($0.03 \pm 0.39$ nmol l$^{-1}$ per degree, $p = 0.14$) [12]. Vitamin D deficiency (low serum 25(OH)D) is very common in western dwelling South Asian populations, including very severe deficiency (less than 12.5 nmol l$^{-1}$) in 27–60% of individuals [13]. Although seasonal variation in vitamin D is found, a study from Birmingham, UK showed that 94% were vitamin D deficient in winter and 82% in summer [14]. Women tended to have lower vitamin D levels than men in summer but not in winter.

### 2.2. Obese people are likely to be vitamin D deficient

There is a very strong association between vitamin D deficiency and obesity. A meta-analysis that included data from 13 209 individuals reported an odds ratio of 3.43 (95% CI 2.33–5.06) for the association between vitamin D deficiency (varying definitions) and obesity [15]. The relationship between serum vitamin D levels and body weight exists across the entire range, i.e. it is not just extreme obesity that is linked with lower vitamin D [16] (figure 1). Thus, ethnicity and increased body weight both independently increase risk of vitamin D deficiency (figure 2). The mechanism for the relationship between obesity and vitamin D deficiency is incompletely understood. It may partly reflect sequestration of vitamin D (which is fat soluble) in body fat but it has been shown that obesity in mice markedly reduces hepatic 25-hydroxylation of vitamin D required for the generation of 25(OH)D [17].

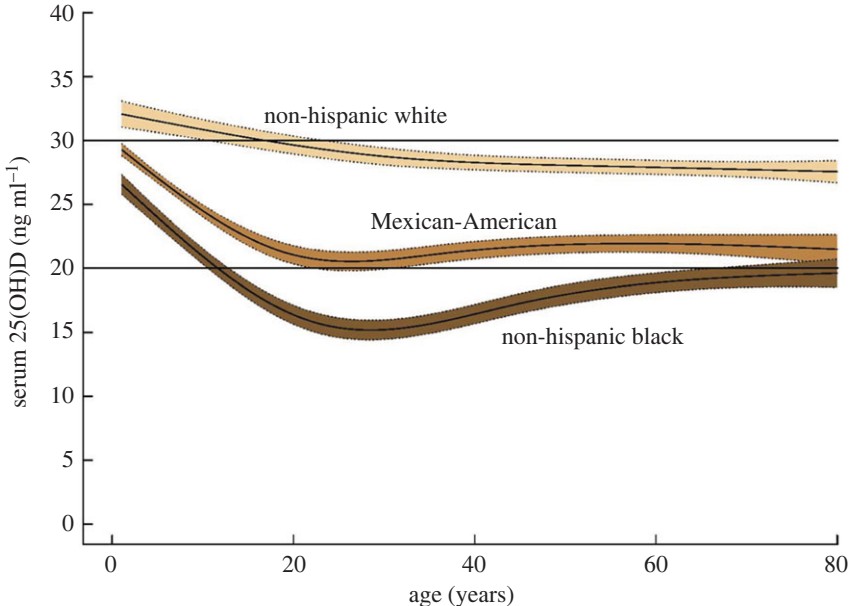

**Figure 2.** Estimated mean serum 25-hydroxyvitamin D (25(OH)D) level, with 95% confidence interval, of greater than or equal to 1-year-old US residents ($n = 31\,934$), by skin colour, over the normal range of body weights from the National Health and Nutrition Examination Survey, 2003–2010. From [16] with permission.

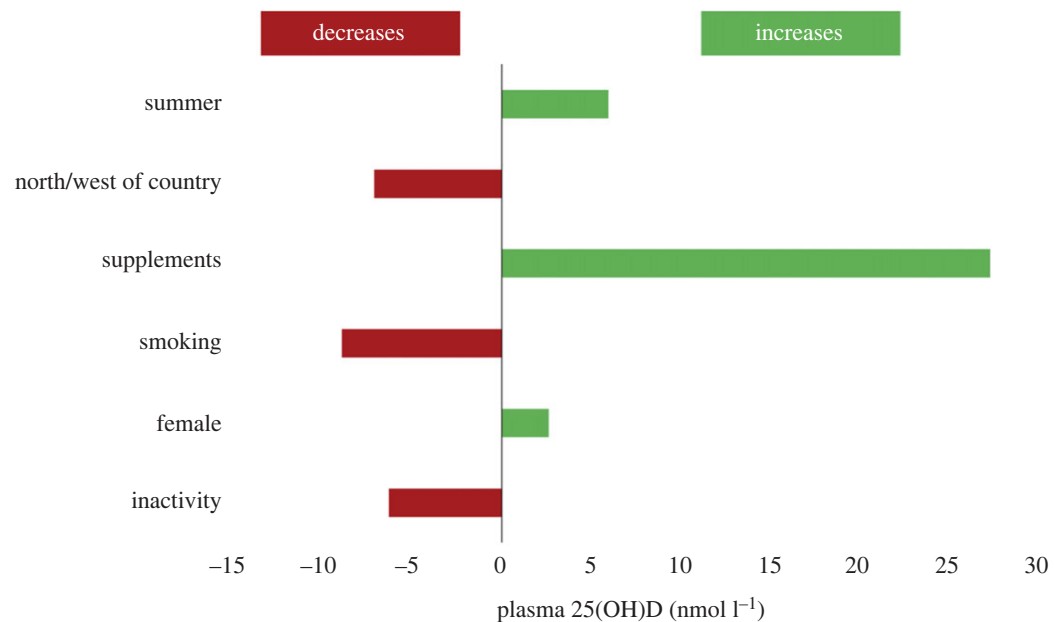

**Figure 3.** Impact of various factors, including supplementation, on blood vitamin D levels in 5382 community-dwelling Irish adults greater than or equal to 50 years. From [19] with permission.

## 2.3. Vitamin D deficiency is common in older people, particularly in those who are institutionalized

Vitamin D synthesis in the skin in response to UVB falls by at least 50% in elderly people compared with young adults [18]. The Irish Longitudinal Study on Aging (TILDA) is a prospective study of community-dwelling adults aged 50 years and older in Ireland and includes 5382 with vitamin D measurement. During the winter, 21.3% of adults aged over 55 years were vitamin D deficient (less than 30 nmol $l^{-1}$). The deficiency was even more prevalent among those aged 80–84 years (29.6%) and those aged over 85 years (46.6%). In winter, only 9.4% of those aged 55+ and 11.5% of those aged 70+ reported taking a vitamin D supplement. A much higher proportion of women (14.6%) than men (3.8%) took a supplement. Taking a supplement was the strongest factor determining vitamin D level [19] (figure 3).

Being institutionalized is an additional risk factor for vitamin D deficiency. A case-control study of 445 institutionalized, osteoporotic women from nine countries (Australia, Belgium, France, Germany, Hungary, Italy, Poland, Spain and UK) found that, in women not taking vitamin D supplements, the serum 25(OH)D concentration was lower in institutionalized women (56.9 [s.d. = 23.9] nmol l$^{-1}$) than non-institutionalized women (63.2 [s.d. = 22.0] nmol l$^{-1}$; $p < 0.0001$) [20]. Vitamin D deficiency is also common among prison inmates, rising to 90% incidence among longer term (more than 1 year) inmates [21,22].

# 3. Vitamin D deficiency may impact the risk of respiratory viral infection

## 3.1. Impact on risk for infection

Meta-analysis of 25 randomized controlled trials including 10 933 participants studied the impact of vitamin D supplementation on risk of respiratory viral infections. This showed a reduction, from 42.2% to 40.3%, in risk of one or more infections with prior vitamin D supplementation [23]. The reduction in infection rate was greater, from 55.0% down to 40.5%, among those who were vitamin D deficient at baseline. The benefit was seen with regular daily dosing but not with intermittent bolus dosing.

Martineau and colleagues have recently updated this meta-analysis—currently published as a non-peer-reviewed pre-print [24, p. 5].

'For the primary comparison of vitamin D supplementation vs. placebo, the intervention reduced risk of ARI (acute respiratory infection) overall (odds ratio [OR] 0.89, 95% CI 0.81–0.98). No statistically significant effect of vitamin D was seen for any of the sub-groups defined by baseline 25(OH)D concentration. However, protective effects were again seen for trials in which vitamin D was given using a daily dosing regimen (OR 0.75, 95% CI 0.61–0.93); at daily dose equivalents of 400–1000 IU (OR 0.70, 95% CI 0.55–0.89); and for a duration of ≤12 months (OR 0.82, 95% CI 0.72–0.94). Vitamin D did not influence the proportion of participants experiencing at least one serious adverse event (OR 0.94, 95% CI 0.81–1.08).'

It can be concluded from this that any effect of vitamin D deficiency on risk for respiratory viral infections is probably modest. It might be argued that some of the respiratory virus infections included in the meta-analysis may not be seasonal and possibly less likely to be impacted by vitamin D status; however, there were studies with proven influenza as outcome and here too the signal was weak.

The lack of impact on serious adverse events seen in the meta-analysis needs to be treated with caution since the included trials were not designed primarily to study this and many patients included in the trials will have had normal vitamin D status at baseline and/or received bolus rather than daily vitamin D.

## 3.2. Impact on the severity of infection

There is evidence that vitamin D deficiency may have a larger impact on the severity of respiratory viral infections. Vo *et al.* [25] showed that the need for intensive care in 1016 infants hospitalized with bronchiolitis was 22% if vitamin D was less than 20 ng ml$^{-1}$ (50 nmol l$^{-1}$), compared with 12% if vitamin D was greater than 30 ng ml$^{-1}$ (75 nmol l$^{-1}$); ($p = 0.003$). Meta-analysis has shown increased odds (OR 1.52, $p = 0.007$) for hospitalization for respiratory syncytial virus (RSV) bronchiolitis in infants who possess a minor allele for a VDR polymorphism (Fok1-f rs2228570) that lowers the transcriptional activity of the VDR [26].

Recently, a 15-year follow-up study of a cohort of 9548 adults, aged 50–75 at baseline, in Saarland, Germany has shown that baseline Vitamin D insufficiency (25(OH)D 30–50 nmol l$^{-1}$) and deficiency (25(OH)D less than 30 nmol l$^{-1}$) were common (44% and 15%, respectively) [27]. Compared to those with sufficient vitamin D status, people with vitamin D insufficiency and deficiency had increased respiratory mortality, with adjusted hazard ratios (95% CI) of 2.1 (1.3–3.2) and 3.0 (1.8–5.2) overall. Associations remained highly significant (HR 3.04, 95% CI 1.79–5.17; $p < 0.001$) after adjustment for age, sex, season, education, smoking, BMI, physical activity and fish consumption. Overall, 41% (95% CI 20–58%) of respiratory disease mortality was attributable to vitamin D insufficiency or deficiency. An inverse dose–response relationship between respiratory mortality and baseline vitamin D blood concentration was noted up to 75 nmol l$^{-1}$ of 25(OH)D.

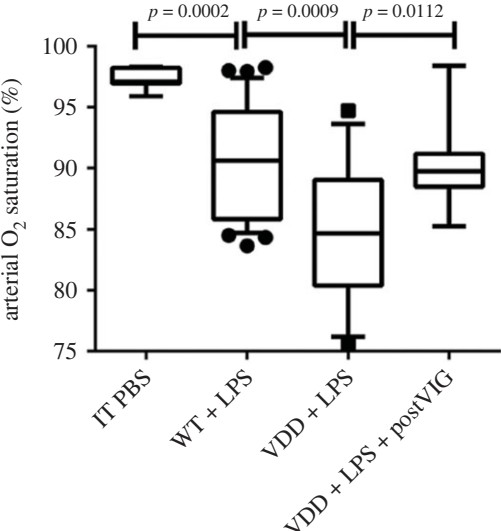

**Figure 4.** Arterial oxygen saturation in wild-type (WT) and vitamin D-deficient mice (VDD) given intratracheal lipopolysaccharide followed by 1500 IU of intra-peritoneal liquid cholecalciferol (Vigantol, VIG) rescue therapy 6 h post-injury. From [44] with permission.

# 4. A molecular and cellular explanation of how vitamin D sufficiency may protect against COVID-19

## 4.1. Cathelicidin and viral killing

Intracellular receptors (VDR) for 1,25(OH)2D are expressed by most immune cells including activated T cells, B cells, dendritic cells and macrophages [28]. Vitamin D is important for the killing of phagocytozed bacteria, including *Mycobacterium tuberculosis* [29,30] and *Escherichia coli* [31], by macrophages. An important part of this bactericidal effect relates to VDR-mediated induction by 1,25(OH)$_2$D of cathelicidin, a cationic bactericidal peptide [32]. Cathelicidin (LL-37) can be produced not only by macrophages but also by epithelial cells and has anti-viral activity, particularly against viruses that, like SARS-CoV-2, are enveloped [33]. Vitamin D induces an anti-viral effect against rhinovirus in cultured respiratory epithelial cells [34], an effect that can also be demonstrated by the addition of exogenous LL-37 [35]. An effect of LL-37 against influenza has also been shown [36].

## 4.2. Modulation of cytokine response and experimental lung damage

The actions of vitamin D on macrophage defence against viral pathogens have shown a predominant impact on cytokine response rather than on viral killing [37]. Some of the work has focused on dengue fever, a viral infection associated with very marked cytokine activation [38,39]. Vitamin D deficiency correlates with increased cytokine generation by the dengue virus *in vitro* although paradoxically one study has shown reduced risk of septic shock in vitamin D-deficient patients with dengue fever [40]. Consistent suppression of the inflammatory cytokine response to pathogens has been shown by vitamin D, in macrophages, in T cells and in various animal models of pneumonia and pneumonitis [41–43].

Vitamin D-deficient mice develop more severe lung injury in response to intratracheal injection of bacterial lipopolysaccharide. These effects were ameliorated by intra-peritoneal injection of cholecalciferol [44] (figure 4).

Studies by Kong *et al.* [42], also in mice, showed that removal of the regulatory effect of vitamin D results in several-fold increase in the inflammatory cytokine (IL-6) response to intra-peritoneal lipopolysaccharide (figure 5). This effect is blocked by an angiotensin-2 inhibitor L1-10. Increased angiotensin-2 activity as a consequence of the interaction between SARS-CoV-2 and its receptor, ACE2, is thought to be central to the hyper-inflammatory lung damage that characterizes severe COVID-19 [45].

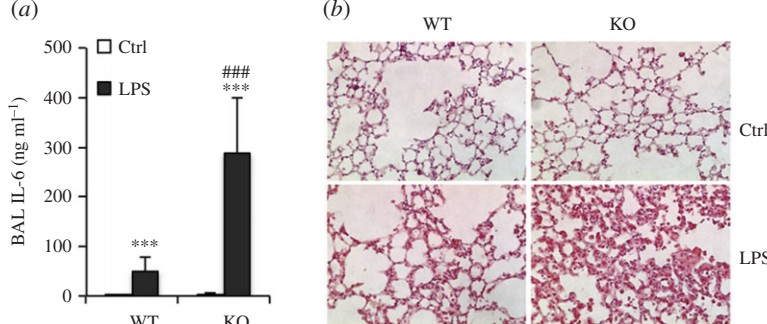

**Figure 5.** (*a*) Blockade of angiotensin II signalling ameliorates lung injury in vitamin D receptor-null mice. Wild-type (WT) and VDR knock-out (KO) mice were pretreated with saline or L1–10 (angiotensin II antagonist) for two weeks followed by LPS challenge. Interleukin-6 (IL-6) levels in the broncho-alveolar lavage (BAL) fluid. ***, $p < 0.001$ versus LPS-treated WT; ###, $p < 0.001$ versus LPS-treated KO; $n = 5–6$ in each genotype. Ctrl, control. IL-6 levels were approximately fivefold increased in vitamin D receptor knock-out mice. From [42] with permission. (*b*) VitD receptor (VDR) inactivation leads to severe acute lung injury after lipopolysaccharide (LPS) challenge. Wild-type (WT) and VDR knock-out (KO) mice were treated with saline or LPS (20 mg kg$^{-1}$, intra-peritoneal injection). The lungs in animals lacking vitamin D receptor (KO) show a marked increase in lung congestion with inflammatory cells following lipopolysaccharide (LPS) challenge. From [42] with permission.

## 4.3. Vitamin D and T cells—and the impact of gender

Vitamin D also suppresses pro-inflammatory cytokines IL17 and interferon gamma and increases the production of the anti-inflammatory cytokine interleukin10 by CD4+ T cells, effects that are much greater in T cells from women than from men. Similarly, anti-CD3- and anti-CD28-stimulated peripheral blood mononuclear cells from men generated substantially fewer regulatory CD4+CD25+ FoxP3+ T lymphocytes in response to vitamin D compared with cells from women, but this gender difference disappeared when oestradiol was added [46]. This could be relevant to the greater severity of COVID-19 reported in men.

Severe COVID-19 is accompanied by exacerbated Th1 responses, which are suspected to contribute towards pathogenic hyper-inflammation. A study, currently reported as a pre-print, reported single-cell RNA sequence datasets from T cells obtained from broncho-alveolar fluid (BALF) [47]. BALF CD4+ T cells from patients with COVID-19 were skewed towards Th1 as opposed to Th2 or Th17 lineages and this was associated with approximately fourfold reduction in the anti-inflammatory cytokine IL10. In a study of 2279 'genesets', the vitamin D-repressed geneset was in the top 1% of all genesets distinguishing COVID-19 patient from healthy Th cells. Overlap was noted between genes that were modulated by vitamin D and those modulated by corticosteroids.

## 5. Latitude associations with COVID-19 mortality imply seasonality and a plausible link with vitamin D

When analysed in May 2020, COVID-19 mortality by country showed a significant association with northern latitude that remained significant after adjustment for age, country population density and pollution. Much of the association between latitude and COVID-19 mortality by country was accounted for by differing age of populations but, adjusting for per cent of population greater than or equal to 65 years left a significant relationship between latitude and COVID-19 mortality ($p = 0.031$) with an estimated 4.4% increase in mortality for each 1° latitude north of 28° North [48]. An association between northerly latitude and mortality was also noted around the same time among African–Americans across the USA [49].

These latitude associations will of course change as we go through the seasons. They imply an effect of ultraviolet light and this has been supported by a further study showing an association between UVB exposure and COVID-19 mortality across 152 countries [50]. This could be mediated by a direct effect of viral killing by UVB in the environment but is also compatible with an effect mediated by vitamin D— the latter would of course fit with the link between COVID-9 severity and ethnicity, whereas the former would not. Either effect would imply seasonality. SARS-CoV-2 is a coronavirus and other coronaviruses along with influenza and RSV, are the classic seasonal respiratory viruses [51] (figure 6).

| month | June | July | Aug | Sep | Oct | Nov | Dec | Jan | Feb | Mar | Apr | May |
|---|---|---|---|---|---|---|---|---|---|---|---|---|
| winter virus | | | | | | influenza virus | | | | | | |
| | | | | | | | HCoV | | | | | |
| | | | | | RSV | | | | | | | |
| all-year virus | adenovirus/HBoV | | | | | | | | | | | |
| type-specific | PIV3 | | PIV1 | | | | | | | | | |
| spring | hMPV | | | | | | | | | | | |
| spring/autumn | rhinovirus | | | | | | | | | | | |
| summer virus | non-rhinovirus enteroviruses | | | | | | | | | | | |

**Figure 6.** Seasonality of respiratory virus infection in temperate regions. Respiratory viruses are classified in three groups according to their seasonal epidemics. Influenza virus, human coronavirus (HCoV) (such as strains OC43, HKU1, 229E and NL63), and human respiratory syncytial virus (RSV) show peaks in winter (winter viruses). Adenovirus, human bocavirus (HBoV), parainfluenza virus (PIV), human metapneumovirus (hMPV) and rhinovirus can be detected throughout the year (all-year viruses). From [51] with permission.

Although the World Health Organization has previously suggested that SARS-CoV-2 may not be a seasonal virus this seems to have been based on reported infection rates rather than on mortality data—the latter arguably demonstrate more convincing seasonality and the consensus is shifting towards the expectation that SARS-CoV-2, like other coronaviruses, is proving seasonal [52]. A further detailed analysis has shown that UV is probably the major determinant of COVID-19 seasonality whereas, contrary to common predictions, warmer temperature positively affects COVID-19 growth rates and humidity has little additional effect [53].

# 6. Preliminary evidence of associations between vitamin D status and COVID-19

## 6.1. Studies based on historical vitamin D measurements

### 6.1.1. By country

Correlations have been shown between the historic prevalence of vitamin D deficiency and COVID-19 mortality per million by country. This has been shown for European countries [54] (figure 7). The surprisingly high vitamin D levels in Scandinavian countries are thought to reflect their strong promotion of vitamin D fortification and supplementation [55]. Vitamin D deficiency (serum 25-hydroxyvitamin D (25(OH)D) $< 50$ nmol l$^{-1}$ or 20 ng ml$^{-1}$) occurs in less than 20% of the population in Northern Europe, in 30–60% in Western, Southern and Eastern Europe and up to 80% in Middle East countries. Severe deficiency (serum 25(OH)D $< 30$ nmol l$^{-1}$ or 12 ng ml$^{-1}$) is found in more than 10% of Europeans [56].

### 6.1.2. By individuals

Data from 348 598 UK Biobank participants have been used to correlate historical vitamin D levels checked between 2006 and 2010 with risk for COVID-19 positivity ($n = 449$) [57]. This included 656 with inpatient-confirmed COVID-19 and 203 deaths from COVID-19. Univariate analysis showed a significantly lower median vitamin D in those testing positive (43.8 nmol l$^{-1}$, IQR 28.7–61.6) than in those without COVID-19 (47.2, IQR 32.7–62.7; $p < 0.01$). Significance was lost on multivariable analysis but the data were substantially changed in a corrigendum that makes this difficult to interpret. The greater than 10-year-old age of the vitamin D levels is of course a major defect in this study that the authors acknowledge. Hastie *et al.* [58] have followed this up with an updated report based on 341 484 UK Biobank participants—again with vitamin D levels checked between 2006 and 2010. 25(OH)D concentration was associated with severe COVID-19 infection and mortality univariably (mortality per 10 nmol l$^{-1}$ 25(OH)D HR 0.92; 95% CI 0.86–0.98; $p = 0.016$), but again significance was lost after adjustment for confounders (mortality per 10 nmol l$^{-1}$ 25(OH)D HR 0.98; 95% CI = 0.91–1.06;

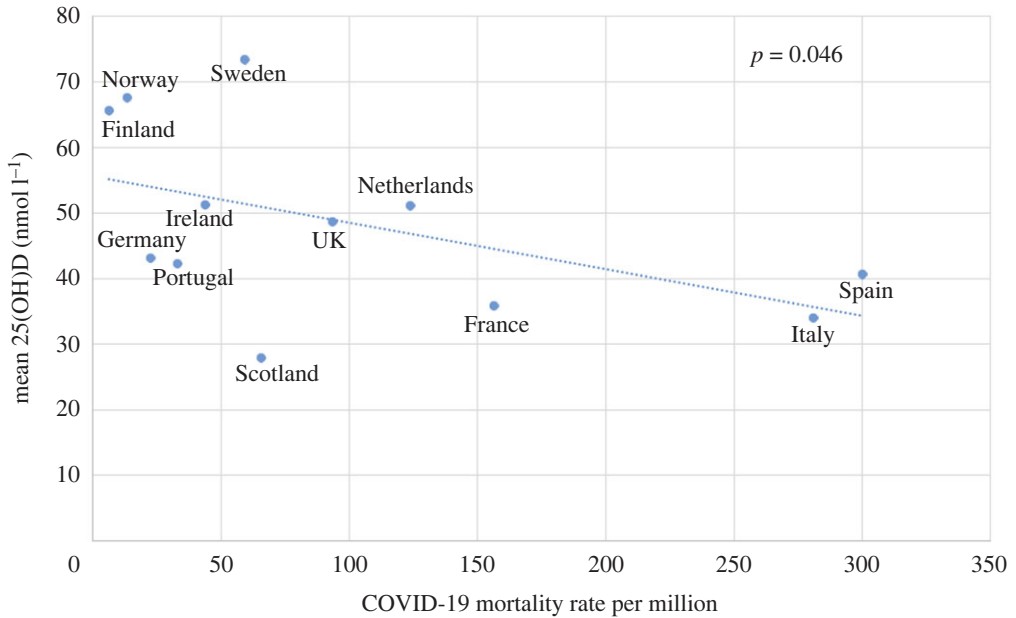

**Figure 7.** Historic prevalence of vitamin D deficiency in European countries compared with mortality per million from COVID-19 by April 2020. From [54] with permission.

$p = 0.696$). It should be noted that the univariate analysis did show a dose response: $25(OH)D < 25 \, \text{mol} \, l^{-1}$ HR for COVID-19 mortality 1.61 (1.14–2.27) $p = 0.007$; $25(OH)D < 50 \, \text{nmol} \, l^{-1}$ 1.29 (0.97–1.72) $p = 0.076$; and a similar dose–response for COVID-19 hospitalization: $25(OH)D < 25 \, \text{nmol} \, l^{-1}$ IRR 1.56 (1.28–1.90) $p < 0.001$; $25(OH)D < 50 \, \text{nmol} \, l^{-1}$ 1.33 (1.14–1.56) < 0.001. Although significance was lost on multivariable analysis this adjusted for 13 criteria that included BMI and ethnicity, both of which are probably causally associated with vitamin D deficiency so this adjustment is very likely to have diminished any effect of vitamin D status.

A separate study has been performed using the UK Biobank by Li *et al.* [59]. This was based on 7898 participants who had been tested for SARS-CoV-2, including 1596 testing positive and 1020 hospitalized. Another 399 COVID-19 cases were identified from the death registry. Again, non-associations were found between historical vitamin D levels and COVID-related outcomes on multivariable analysis. However, they performed a separate analysis that looked at estimated UVB exposure for each individual in the 135 days prior to diagnosis and used this to derive a measure of likely vitamin D synthesis 'vitD-UVB' over the months preceding diagnosis. They validated this by showing that a similar analysis at time of sampling correlated strongly with serum vitamin D levels. VitD-UVB dose was not associated with COVID-19 infection risk but was strongly and inversely associated with the hospitalization and death from COVID-19 in univariable and multivariable models (OR = 0.98, 95% CI: 0.97–0.99, $p$-value $< 2 \times 10^{-16}$). Among COVID-19 positive cases vitD-UVB doses were more than 50% lower in those who died (43.09 kJ $M^{-2}$ (31.89–74.10) median (IQR)) compared with those who did not require hospital admission (90.80 kJ $M^{-2}$ (67.43–98.95)). These data support the hypothesis that vitamin D status impacts on COVID-19 severity more than on risk of COVID-19 infection.

A population-based study among 4.6 million members of a Health Services provider in Israel showed that vitamin D levels estimated between 2010 and 2019 correlated strongly with COVID-19 incidence [60]. They found that vitamin D deficiency (less than 50 nmol $l^{-1}$), and particularly severe vitamin D deficiency (less than 30 nmol $l^{-1}$) was much more prevalent among Ultra-orthodox and Arabs. On multivariable analysis with adjustment for ethnicity severe vitamin D deficiency was associated with an OR 1.275 (95% CI 1.199–1.355; $p < 0.001$) for COVID-19 infection.

## 6.2. Studies based on recent vitamin D measurements prior to COVID-19

A study of 7807 Health Service members in Israel who had serum 25(OH)D levels previously checked (date not specified) included 782 (10.1%) who tested positive for COVID-19 and 7025 (89.9%) who tested negative. Multivariate analysis showed that 'low vitamin D' (less than 30 ng ml$^{-1}$ [less than

75 nmol l$^{-1}$]) had an OR of 1.45 (1.08–1.95; $p < 0.001$) for COVID-19 positivity and an OR of 1.95 (0.98–4.845; $p = 0.061$) for hospitalization [61].

A retrospective cohort from Chicago included 4314 patients tested for COVID-19, all of whom had a vitamin D level checked in the year before testing [62]. In multivariate analysis that adjusted for age and ethnicity, being likely vitamin D deficient (previous deficient level [25(OH)D3 < 20 ng ml$^{-1}$, equivalent to less than 50 nmol l$^{-1}$] and lack of subsequent supplementation) associated with increased risk of testing positive for COVID-19 (RR 1.77, $p < 0.02$).

A study of 191 779 patients from across all US states used data from tests performed at a national clinical laboratory [63]. All patients had SARS-CoV-2 test results (9.3% positive) and all had 25(OH)D results available from the preceding 12 months. A strong association was found between lower 25(OH)D levels (seasonally adjusted for date of sampling) and increased rate of SARS-CoV-2 positivity. This remained after adjustment for sex, age, latitude and ethnicity (adjusted odds ratio 0.984 per ng ml$^{-1}$ increment of 25(OH)D, 95% CI 0.983–0.986; $p < 0.001$). The dose–response did not appear to plateau until 25(OH)D levels approached 55 ng ml$^{-1}$ (137.5 nmol l$^{-1}$). This study did not include data on COVID-19 severity.

## 6.3. Studies based on vitamin D measurements during COVID-19

1,25(OH)$_2$D is present in serum only at very low levels and with a short half-life. Most studies, therefore, measure serum levels of 25-hydroxyvitamin D (25(OH)D). This is predominantly protein bound, mainly to the vitamin D binding protein but also to albumin. Concentrations of both vitamin D binding protein and albumin tend to fall in illness, as a negative acute phase response. This tends to lower circulating total vitamin D but releases free 25(OH)D that can be taken up by cells of the immune system and epithelial cells. The consequences of changes in serum 25(OH)D during illness are, therefore, complex and difficult to interpret [8].

### 6.3.1. Associations between serum 25(OH)D levels and risk of COVID-19 infection

A study from Italy has reported serum vitamin D levels taken with seven weeks of SARS-CoV-2 PCR testing in 107 patients—mostly within 3 days of test [64]. People ($n = 27$) testing positive for SARS-CoV-2 had lower 25(OH)D serum concentrations, median 11.1 ng ml$^{-1}$ (IQR 8.2–21.0) compared with those testing negative ($n = 80$), 25(OH)D median 24.6 ng ml$^{-1}$ (IQR 8.9–30.5) ($p = 0.004$).

Conversely, another retrospective study from Italy of 347 hospital patients found no significant difference in 25(OH)D serum levels between those who were COVID-19 positive ($n = 128$, 25(OH)D 21.8 ± 16.1 ng ml$^{-1}$) and those who were COVID-19 negative ($n = 219$, 25(OH)D 22.8 ± 14.0 ng ml$^{-1}$) [65].

Nutritional status was assessed in 50 adults hospitalized for COVID-19 in South Korea and compared with 150 age- and sex-matched controls attending for annual checks [66]. Vitamin D deficiency (less than or equal to 20 ng ml$^{-1}$ [less than or equal to 50 nmol l$^{-1}$]) was present in 37/50 (74%) admitted COVID-19 patients compared with 65/150 (43.3%) controls ($p = 0.003$). Severe vitamin D deficiency (less than or equal to 10 ng ml$^{-1}$ [less than or equal to 25 nmol l$^{-1}$]) was present in 12/50 (24%) COVID-19 patients compared with 11/150 (7.3%) controls, ($p < 0.001$).

A study of 392 hospital staff from Birmingham, UK has shown that being of black, Asian or ethnic minority (BAME) or vitamin D deficient (less than 30 nmol l$^{-1}$) were independent predictors of testing positive for SARS-CoV-2 antibodies [67]. Overall 72% of those with vitamin D deficiency tested positive compared with 51% of those without vitamin D deficiency ($p = 0.003$). On multivariate analysis that adjusted for age, gender, BMI, ethnicity, comorbidities and job role, vitamin D deficiency conferred an odds ratio of 2.6 (95% CI 1.41–4.80; $p = 0.002$) for seroconversion. Vitamin D levels were checked when subjects were well or convalescent so are unlikely to have been affected by a negative acute phase response.

### 6.3.2. Associations between serum 25(OH)D levels and COVID-19 severity

A study of 42 consecutive adult patients admitted to intensive care in Bari, Italy reported a mortality by day 10 of 50% among 10 patients with severe vitamin D deficiency (less than 10 ng ml$^{-1}$ [less than 25 nmol/]) compared with 5% mortality among 32 with greater than or equal to 10 ng ml$^{-1}$ [25 nmol l$^{-1}$], $p = 0.019$ [68]. An audit of 134 patients admitted to hospital with COVID-19 in Newcastle UK has reported that only 19% of ITU patients had 25(OH)D levels greater than 50 nmol l$^{-1}$ compared with 39.1% of non-ITU patients ($p = 0.02$) [69].

Many of the studies reported to date have been retrospective but a prospective cohort study has recently been reported from Wrexham Park Hospital, UK [70]. This included all 105 emergency

admissions greater than or equal to 65 years, of whom 70 proved COVID-19 positive and 35 COVID-19 negative. No patients were admitted to intensive care but among patients with vitamin D deficiency in the COVID-19-positive group, there was a higher incidence of non-invasive ventilatory support and high dependency unit admission (30.77% versus 9.68%) ($p = 0.042$). Vitamin D levels in the COVID-19-positive group were also significantly lower compared with those in the COVID-19-negative group (median 27.00 nmol l$^{-1}$ versus 52.00 nmol l$^{-1}$; $p = 0.0008$).

A study from Heidelberg, Germany has reported on 185 consecutive SARS-CoV-2 positive patients on a prospective non-interventional register [71]. Serum 25(OH)D was measured retrospectively in samples collected and frozen at time of admission to the study. The decision to admit to hospital was based on SpO$_2$ less than or equal to 93% and overall performance status without knowledge of vitamin D status. Vitamin D deficiency (defined as less than 12 ng ml$^{-1}$ = less than 30 nmol l$^{-1}$) was present in 31% of the 93 admitted compared with 13% of the 92 managed as outpatients ($p = 0.004$). Among those admitted, 62% of those with vitamin D deficiency required high flow oxygen or mechanical ventilation compared with 27% of those with vitamin D sufficiency ($p = 0.004$). After adjustment for age, gender and comorbidities, vitamin D deficiency was strongly associated with combined risk for mechanical ventilation and/or death: HR (95%CI) 6.12 (2.79–13.42; $p < 0.001$ across the entire cohort; HR4.65 (2.11–10.25; $p < 0.001$) in inpatient subgroup. Vitamin D deficiency after adjustment for age, gender and comorbidities was also strongly associated with increased risk for death: HR 14.73 (4.16–52.19; $p < 0.001$) across the whole cohort; HR 11.51 (3.24–40.92; $p < 0.001$) in inpatient subgroup. The authors noted that when applying a higher cut-off for vitamin D deficiency (20 ng ml$^{-1}$ = 50 nmol l$^{-1}$), by which definition 64% of their patient population were deficient, the associations between low vitamin D and severity of COVID-19 were maintained across the whole cohort: vitamin D less than 20 ng ml$^{-1}$ (50 nmol l$^{-1}$) had HR 11.27 (1.48–85.55; $p = 0.02$) for risk of death.

A study of 109 hospitalized patients from multiple centres in Austria only had admission sera available for analysis in 37 of the patients and did not find significant correlation between vitamin D deficiency on admission and outcome; however, when vitamin D levels were re-checked at eight weeks follow-up they were significantly lower in those who had suffered severe disease ($p < 0.05$) [72]. Parathyroid hormone levels, which increase with vitamin D deficiency, and were available in all patients at follow-up, were higher in those who needed intensive care or prolonged oxygen treatment ($p < 0.01$).

A cross-sectional analysis of a registry of 235 hospital-based patients with COVID-19 from Tehran, Iran showed that, among those over 40 years old, mortality was reduced (9.7% compared with 20%; $p = 0.04$) if serum vitamin D was 30 ng ml$^{-1}$ (75 nmol l$^{-1}$) or higher [73].

## 6.4. Studies based on use of supplements during COVID-19

A questionnaire-based study was performed in Italian patients with Parkinson's disease ($n = 1486$) and their family members ('controls' $n = 1207$) [74]. One hundred and five (7.1%) patients and 92 (7.6%) family members had confirmed or probable COVID-19. Vitamin D supplements had been taken by 13/105 (12.4%) COVID-19 cases compared with 316/1381 (22.9%) unaffected—after age adjustment OR 0.56 (95% CI 0.32–0.99; $p = 0.048$) for COVID-19 infection in those taking vitamin D supplements.

A recent small study from a nursing home in Angers, France showed reduced mortality from COVID-19 among elderly patients (average age in both groups ca 87 years) who received an oral bolus of vitamin D 80 000 IU either in the week following suspicion or diagnosis of COVID-19 or in the previous month (10/57 = 17.5% mortality) compared with a smaller group who did not receive vitamin D (5/9 = 55.6% mortality; $p = 0.023$) [75]. Although the groups were well-matched for age, the study was retrospective and not randomized.

## 7. Randomized controlled trials of supplementary vitamin D in COVID-19

Reviews of the evidence for a role of vitamin D supplementation in protecting against COVID-19 have pointed to the lack of randomized controlled trials. It is important though to consider the practicalities of performing such a trial. It should preferably be done in the community since giving vitamin D to deficient patients after they are already ill with COVID-19 may be too late. It should also be done in people who are deficient at baseline since supplementing people who already have adequate vitamin D status is unlikely to benefit. This implies recruiting people identified as vitamin D deficient and then inviting them to consent to randomization with the possibility of receiving placebo. While this can readily be accepted for a trial of a novel therapeutic with unknown efficacy and safety it is another matter altogether to run a placebo-controlled trial for a vitamin that is known to be essential for health.

This may possibly explain why, until recently, only one such trial, based in Tehran, was registered on clintrials.gov: NCT04386850. A UK-based trial (CORONAVIT NCT04579640) is now underway comparing higher dosage (800 IU/day or 3200 IU/day) with standard UK government recommendation (400 IU/day). It should be noted though that these trials will not complete until Spring 2021 at the earliest.

An open-label randomized trial of calcifediol (25(OH)D3) in hospitalized patients with COVID-19 has recently (August 2020) been reported from Cordoba, Spain [76]. Patients who were being treated with hydroxychloroquine and azithromycin were also randomized 1 : 2 to either standard care or standard care plus high-dose oral calcifediol—532 µg on admission, 266 µg on days 3 and 7, then weekly till discharge or admission to intensive care. The primary endpoint was admission to intensive care which was required in only 1/50 patients receiving vitamin D versus 13/26 who did not receive vitamin D. This difference was highly significant even after adjustment for hypertension and diabetes (OR 0.03; 95% CI 0.003–0.25). The use of calcifediol is potentially very important since this is already 25-hydroxylated and available for uptake by immune cells which possess the 1-hydroxylase necessary to complete the activation. This could substantially bring forwards the availability of the vitamin D to the immune system by bypassing the need to wait for 25-hydroxylation to occur in the liver.

The relative lack of randomized trial data, particularly with respect of vitamin D supplementation in the community, does not necessarily imply that vitamin D supplementation is not justified on pragmatic grounds. A group of doctors from Slovenia [77] have pointed out that the European Centre for Disease Prevention and Control recommends that

'Public health authorities should recognise that extra-scientific factors (e.g. feasibility of implementing scientific advice, time pressure, socio-political factors, institutional factors, economic interests, pressure from neighbouring countries, etc.) are inherent to the decision-making process. These factors will also influence the implementation of any proposed response measures. Decisions should, therefore, always be evidence informed, but they will very rarely be purely evidence based' [78, p. 2].

This makes the case for recommending pragmatic vitamin D supplementation on the basis of indirect evidence much as has been done for social distancing and wearing of masks.

# 8. Identifying the appropriate vitamin D target blood levels and supplement dosing

## 8.1. Identifying the optimum target blood level

There is remarkable uncertainty about the healthy serum concentration for vitamin D. Recommendations by authoritative bodies range from greater than 25 nmol l$^{-1}$ (UK Scientific Advisory Committee on Nutrition, SACN), through greater than 50 nmol l$^{-1}$ (US Institute of Medicine, now the US National Academy of Medicine) to greater than 75 nmol l$^{-1}$ (US Endocrine Society) [5]. The vitamin D intake required for health depends very much on the target serum concentration. We have recently reviewed the evidence underlying the varying definitions of sufficiency and could find no published evidence to support the UK SACN definition of 25 nmol l$^{-1}$ for sufficiency [79]. Evidence supporting 50 nmol l$^{-1}$ as a recommended blood vitamin 25(OH)D level was provided by studies of parathyroid hormone changes in response to vitamin D supplementation in people with different baseline levels of vitamin D [80]. People with baseline vitamin D greater than or equal to 50 nmol l$^{-1}$ showed no significant fall in parathyroid hormone. Large population-based studies correlating all-cause mortality with serum vitamin D concentration also support an optimal level of at least 50 nmol l$^{-1}$ [79,81,82] (figure 8). It has been suggested that there might be a U-shaped curve to the dose–response with a potentially harmful effect at very high vitamin D levels [83] but this was not confirmed in the recent UK Biobank study [82]. Amrein *et al.* [84] have recently reviewed the topic and again came to the conclusion that a target serum concentration of 20 ng ml$^{-1}$ (50 nmol l$^{-1}$) is an appropriate minimum level for sufficiency.

## 8.2. The importance of a regular daily dose rather than bolus supplementation

There is evidence from randomized controlled trials of vitamin D supplementation with respiratory virus infections as endpoint that intermittent high-dose boluses are ineffective even though they yield satisfactory blood levels of 25(OH) vitamin D. Thus in their 2017 meta-analysis, Martineau *et al.* [23, p. 1] noted that 'in subgroup analysis, protective effects were seen in those receiving daily or weekly vitamin D without additional bolus doses (adjusted odds ratio 0.81, 0.72–0.91) but not in those receiving one or more bolus doses (adjusted odds ratio 0.97, 0.86–1.10; p for interaction = 0.05).' In this

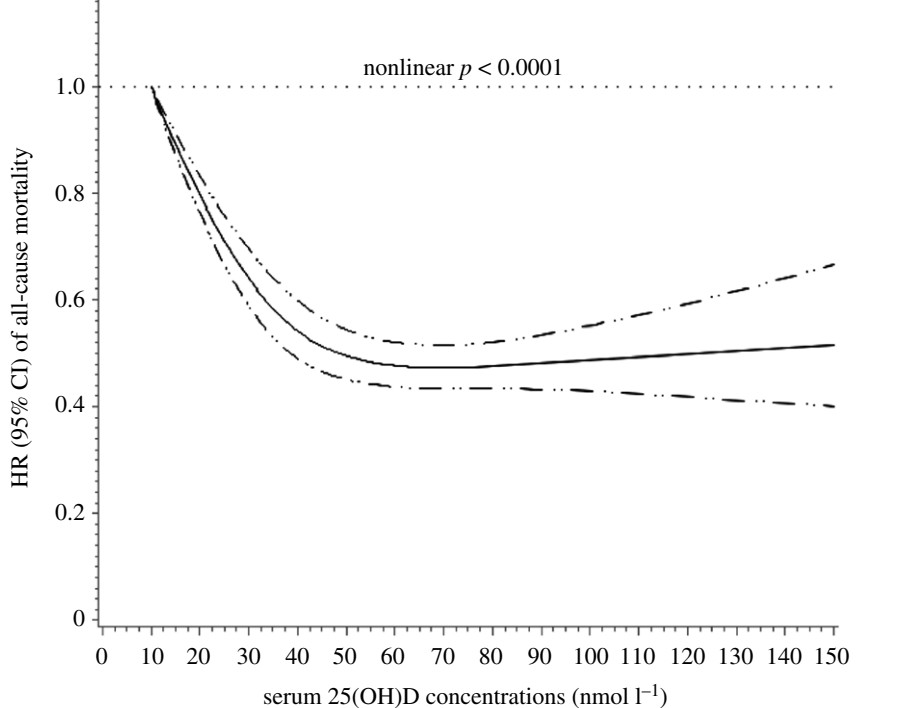

**Figure 8.** Vitamin D status and risk of all-cause mortality in a cohort of 365 530 individuals with median follow-up 8.9 years from the UK Biobank. From [82] with permission.

meta-analysis, data were not presented separately for daily versus weekly supplementation. In the updating of this meta-analysis recently published as a pre-print, Joliffe *et al.* [24] show significant protection against risk of respiratory infection with daily supplementation (OR 0.75 95% CI 0.61–0.93) but not with weekly or monthly bolus.

The lack of benefit from intermittent bolus vitamin D supplementation has also been shown in other contexts. Martineau and colleagues have recently reported lack of efficacy of supplementation with vitamin D 14 000 IU as a weekly dose over 3 years in 8851 Mongolian schoolchildren. Infection with tuberculosis as defined by a positive QuantiFERON-TB test was the primary outcome with the incidence of clinical tuberculosis and ARIs as secondary outcomes. The weekly vitamin D bolus had no significant impact on any of these outcomes [85]. This contrasts with a previous study in 247 Mongolian schoolchildren in which daily ingestion of milk fortified with 300 IU of vitamin D3 resulted in a 50% reduction in participant-reported ARIs during the winter [86]. Similarly, a large placebo-controlled study of bolus supplementation with 100 000 IU every three months in 3046 infants in Afghanistan showed no reduction in risk for pneumonia, and even some increased risk for pneumonia recurrence in the treated group [87].

A possible explanation for the poor results with high-dose bolus may be that the high blood vitamin D levels that will transiently arise with this approach could be harmful. This would be in keeping with the 'U-shaped' curve seen in some all-cause mortality data discussed earlier. It is also relevant that a randomized controlled trial in 311 community-dwelling adults aged 55–70 showed that regular dosing of vitamin D3 at 4000 IU/day or 10 000 IU/day resulted in reduced bone mineral density compared with lower daily dosing at 400 IU/day [88].

A further explanation for poor results with bolus vitamin D could be that this leads to the induction of increased levels of 24-hydroxylase activity leading to inactivation of vitamin D3 via the formation of 24,25 $(OH)_2D$. This effect has been shown to last for at least 28 days after bolus dosing [89].

It seems likely that high doses, whether as high regular daily doses (4000 IU/day or greater) or as intermittent high bolus doses, probably confer no benefit.

## 8.3. Identification of the daily supplement dosage needed to achieve optimal vitamin D status

A meta-analysis of 94 cohort studies that included 11 566 supplemented individuals combined with modelling to allow for age and obesity suggests that for adults, supplementation with 800 IU/day

should be adequate, even in obese individuals, for achieving greater than $50 \, \text{nmol} \, \text{l}^{-1}$. However, to achieve greater than $75 \, \text{nmol} \, \text{l}^{-1}$ would typically require supplementation of between 3000 and 4000 IU/day for an obese individual [90].

Regarding safety—SACN (2016) reiterates the upper limits for vitamin D recommended by the European Food Safety Authority (EFSA), of $100 \, \mu\text{g} \, \text{d}^{-1}$ (4000 IU/day) for adults and children aged 11–17 years, $50 \, \mu\text{g} \, \text{d}^{-1}$ (2000 IU/day) for children aged 1–10 years, and $25 \, \mu\text{g} \, \text{d}^{-1}$ (1000 IU) for infants [91].

It seems reasonable to conclude that a target minimum serum vitamin D concentration of $50 \, \text{nmol} \, \text{l}^{-1}$ is appropriate and that a regular daily dose of 800 IU should be sufficient, even in obese individuals [79]. Since vitamin D is widely sold in 1000 IU capsules, and it is agreed that doses of up to 4000 IU/day are safe for all, then a recommendation of 1000 IU/day for all should be safe and sufficient.

# 9. Conclusion

— Evidence linking vitamin D deficiency with COVID-19 severity is circumstantial but considerable— links with ethnicity, obesity, age and institutionalization; latitude association; evidence from experimental models of respiratory pathogens; preliminary reports of associations with COVID-19 severity in hospitalized patients; basic biology studies showing extensive vitamin D impacts on the immune system underlying various anti-viral and anti-inflammatory responses; vitamin D responsive genes altered in lung lymphocytes from COVID-19 patients.
— Vitamin D deficiency is common; particularly, but not solely, in people living well away from the equator and can persist throughout the year in individuals who have little sunlight exposure.
— Vitamin D deficiency is readily preventable by supplementation that is very safe and cheap.
— The current UK definition of vitamin D deficiency (less than $25 \, \text{nmol} \, \text{l}^{-1}$) is low by international standards and evidence from parathyroid hormone status (which rises with vitamin D deficiency) as well as large population studies of all-cause mortality suggests that a target blood level of at least $50 \, \text{nmol} \, \text{l}^{-1}$, as indicated by the American Institute of Medicine and by the European Food Safety Authority, is more appropriate. This would require supplementation with 800 IU/day (not 400 IU/day as currently recommended in UK) to bring most individuals up to the normal range. Growing evidence suggests that regular daily supplementation is more effective than intermittent high-dose bolus.
— Vitamin D supplementation at levels needed to avoid deficiency—e.g. 800–1000 international units ($25 \, \mu\text{g}$) vitamin D3/day—is extremely safe and typically costs no more than 10p per day.
— Randomized placebo-controlled trials of vitamin D in the community in vitamin D-deficient individuals will be difficult to conduct and will not complete until spring 2021 at earliest— although we note the positive results from Cordoba, Spain of an open-labelled randomized trial of 25(OH)D3 (calcifediol) in hospitalized patients.

Key recommendations:

— There seems nothing to lose and potentially much to gain by recommending vitamin D supplementation for all, e.g. at 800–1000 IU/day, making it clear that this is to help ensure immune health and not solely for bone and muscle health.
— This should be mandated for prescription in care homes, prisons and other institutions where people are likely to have been indoors for much of the time during the summer.
— People likely to be currently deficient should consider taking a higher dose, e.g. 4000 IU/day for the first four weeks before reducing to 800 IU–1000 IU/day
— People admitted to hospital with COVID-19 should have their vitamin D status checked and/or supplemented and consideration should be given to testing high-dose calcifediol in the RECOVERY trial.

We feel this matter should be pursued with great urgency. Vitamin D levels in the UK will be falling from October onwards as we head into winter.

Data accessibility. This article does not contain any additional data.
Authors' contributions. J.R. drafted the first version. All authors made substantial contributions to the conception of the report and interpretation of data, contributed to drafting the article and revising it critically for important intellectual content. All have approved the submitted version, and agree to be accountable for all aspects of the work in ensuring that questions related to the accuracy or integrity of any part of the work are appropriately investigated and resolved.

Competing interests. M.H. and D.T. have received speaking honoraria from Thornton Ross. No other competing interests.
Funding. We received no funding for this study.

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
