## [Reviewer comments · Royal Society Open Science]

Review History

Decision letter (RSOS-201912.R0)

Dear Professor Rhodes

The Editors assigned to your paper RSOS-201912 "Vitamin D and COVID-19 – a need for action: Evidence synthesis and recommendations" have reviewed your submission, and would like you to revise the paper in accordance with their comments below. Please note this decision does not guarantee eventual acceptance.

We invite you to respond to the comments supplied below and revise your manuscript. Below the Editor's comments we provide additional requirements. Final acceptance of your manuscript is dependent on these requirements being met. We provide guidance below to help you prepare your revision.

If deemed necessary by the Editors, your manuscript will be sent for external review after receipt of your revised paper.

Please submit your revised manuscript and required files (see below) no later than 21 days from today's (ie 05-Nov-2020) date. Note: the ScholarOne system will 'lock' if submission of the revision is attempted 21 or more days after the deadline. If you do not think you will be able to meet this deadline please contact the editorial office immediately.

on behalf of Professor John Dalton (Associate Editor) and Catrin Pritchard (Subject Editor)
 openscience@royalsociety.org

Editor comments:

The manuscript is very interesting, extremely topical and make an important and convincing argument concerning the importance of Vitamin supplementation to protect against inflammatory disease, including COVID-19. The data regarding the impact of Vitamin D and COVID-19 is new and yet to be fully substantiated by further experiments but this paper collects the data together well enough to support each of their key recommendations.

I feel the paper could be improved by -

- 1 A simpler title such as 'Vitamin D and COVID-19'
2. Fuse sections 1, 2 and 3 under one heading 'Vitamin D is a hormone activated by sunlight' (there are far too many headings)
3. Fuse the sections 4, 5, 6 and 7 under one heading such as 'People with dark skin, that are obese or are elderly are more likely to be Vitamin D deficient'
4. Because the paper is multi-authored it seems that different sections were written by different authors with different styles. The section 8 dealing with Vit D and inflammation become too detailed in recent cellular and molecular immunology which hasn't been fully supported by repeats from different laboratories. I think this could be reduced and written less detailed - possibly use a Table as a place to show the immunological data and the just review this in the text. Also, this section sits strangely at this point in the paper and perhaps could be moved to between section 12 and 13 under the heading 'A molecular and cellular explanation of how Vitamin D may protect against COVID-19'
5. Heading 9 may be better written 'Vitamin D deficiency may impact the risk of respiratory viral infection'
6. Some sections have sentences in 'bold' perhaps to emphasis certain point - but I find this unnecessary and a little distracting.
7. The article is very long and could benefit by being reduced a little, perhaps ~20%.

===PREPARING YOUR MANUSCRIPT===

===PREPARING YOUR REVISION IN SCHOLARONE===

- If you are providing image files for potential cover images, please upload these at this step, and inform the editorial office you have done so. You must hold the copyright to any image provided.
- A copy of your point-by-point response to referees and Editors. This will expedite the preparation of your proof.

- Ensure that your data access statement meets the requirements at <https://royalsociety.org/journals/authors/author-guidelines/#data>. You should ensure that you cite the dataset in your reference list. If you have deposited data etc in the Dryad repository, please include both the 'For publication' link and 'For review' link at this stage.
- If you are requesting an article processing charge waiver, you must select the relevant waiver option (if requesting a discretionary waiver, the form should have been uploaded at Step 3 'File upload' above).
- If you have uploaded ESM files, please ensure you follow the guidance at <https://royalsociety.org/journals/authors/author-guidelines/#supplementary-material> to include a suitable title and informative caption. An example of appropriate titling and captioning may be found at https://figshare.com/articles/Table_S2_from_Is_there_a_trade-off_between_peak_performance_and_performance_breadth_across_temperatures_for_aerobic_scorpions_in_teleost_fishes_/3843624.

Author's Response to Decision Letter for (RSOS-201912.R0)

See Appendix A.

Decision letter (RSOS-201912.R1)

Dear Professor Rhodes,

It is a pleasure to accept your manuscript entitled "Vitamin D and COVID-19 – evidence and recommendations for supplementation" in its current form for publication in Royal Society Open Science.

COVID-19 rapid publication process:

We are taking steps to expedite the publication of research relevant to the pandemic. If you wish, you can opt to have your paper published as soon as it is ready, rather than waiting for it to be published the scheduled Wednesday.

This means your paper will not be included in the weekly media round-up which the Society sends to journalists ahead of publication. However, it will still appear in the COVID-19 Publishing Collection which journalists will be directed to each week (<https://royalsocietypublishing.org/topic/special-collections/novel-coronavirus-outbreak>).

If you wish to have your paper considered for immediate publication, or to discuss further, please notify openscience_proofs@royalsociety.org and press@royalsociety.org when you respond to this email.

on behalf of Professor John Dalton (Associate Editor) and Catrin Pritchard (Subject Editor)
openscience@royalsociety.org

Appendix A

9th November 2020

Re: RSOS-201912 "Vitamin D and COVID-19 – a need for action: Evidence synthesis and recommendations".

Dear Professors Dalton and Pritchard,

Many thanks for your very helpful and positive comments. We have taken all of these on board and modified the paper accordingly – see responses noted below and changes to manuscript in red font.

Editor comments:

The manuscript is very interesting, extremely topical and make an important and convincing argument concerning the importance of Vitamin supplementation to protect against inflammatory disease, including COVID-19. The data regarding the impact of Vitamin D and COVID-19 is new and yet to be fully substantiated by further experiments but this paper collects the data together well enough to support each of their key recommendations.

– many thanks.

I feel the paper could be improved by -

1 A simpler title such as 'Vitamin D and COVID-19'

- agree – but we feel this is a bit too simple, partly since there are now other published reviews with the same or similar titles – we have suggested:

"Vitamin D and COVID-19 – evidence and recommendations for supplementation"

2. Fuse sections 1, 2 and 3 under one heading 'Vitamin D is a hormone activated by sunlight' (there are far too many headings)

- done (but with some sub-headings which we hope you will allow)

3. Fuse the sections 4, 5, 6 and 7 under one heading such as 'People with dark skin, that are obese or are elderly are more likely to be Vitamin D deficient'

- done

4. Because the paper is multi-authored it seems that different sections were written by different authors with different styles. The section 8 dealing with Vit D and inflammation become too detailed in recent cellular and molecular immunology which hasn't been fully supported by repeats from different laboratories. I think this could be reduced and written less detailed - possibly use a Table as a place to show the immunological data and the just review this in the text. Also, this section sits strangely at this point in the paper and perhaps could be moved to between section 12 and 13 under the heading 'A molecular and cellular explanation of how Vitamin D may protect against COVID-19'

- agree, we have trimmed the laboratory immunology section substantially, used the heading you suggest, and moved it later - just before the recent clinical evidence obtained during the pandemic

5. Heading 9 may be better written 'Vitamin D deficiency may impact the risk of respiratory viral infection'

- done

6. Some sections have sentences in 'bold' perhaps to emphasis certain point - but I find this unnecessary and a little distracting.

- done – bold removed

7. The article is very long and could benefit by being reduced a little, perhaps ~20%.

-done – immunology substantially trimmed and we have also trimmed the last section on dosing. Total text now 6619 words.

Very many thanks for your helpful comments which have substantially improved the text. We hope this is now acceptable for publication.

Best wishes,
Yours sincerely,
Jon Rhodes